# Implementation of an IoT Based Radar Sensor Network for Wastewater Management

**DOI:** 10.3390/s19020254

**Published:** 2019-01-10

**Authors:** Adam Drenoyanis, Raad Raad, Ivan Wady, Carmel Krogh

**Affiliations:** 1School of Electrical, Computer and Telecommunications Engineering (SECTE), University of Wollongong (UOW), Wollongong 2522, NSW, Australia; raad@uow.edu.au; 2Shoalhaven Water, Shoalhaven City Council, Nowra 2541, NSW, Australia; ivan.wady@shoalhaven.nsw.gov.au (I.W.); carmel.krogh@shoalhaven.nsw.gov.au (C.K.)

**Keywords:** communications, Internet of Things, IoT, LoRaWAN, LPWAN, sewer pumping stations, wastewater, wastewater collection network, WCN

## Abstract

Critical wastewater events such as sewer main blockages or overflows are often not detected until after the fact. These events can be costly, from both an environmental impact and monetary standpoint. A standalone, portable radar device allowing non-invasive benchmarking of sewer pumping station (SPS) pumps is presented. Further, by configuring and deploying a complete Low Power Wide Area Network (LPWAN), Shoalhaven Water (SW) now has the opportunity to create “Internet of Things” (IoT)-capable devices that offer freedom from the reliance on mobile network providers, whilst avoiding congestion on the existing Supervisory Control and Data Acquisition (SCADA) telemetry backbone. This network infrastructure allows for devices capable of real-time monitoring to alert of any system failures, providing an effective tool to proactively capture the current state of the sewer network between the much larger SPSs. This paper presents novel solutions to improve the current wastewater network management procedures employed by SW. This paper also offers a complete review of wastewater monitoring networks and is one of the first to offer robust testing of Long Range Wide Area Network (LoRaWAN) network capabilities in Australia. The paper also provides a comprehensive summary of the LoRa protocol and all its functions. It was found that a LPWAN, utilising the LoRaWAN protocol and deployed appropriately within a geographic area, can attain maximum transmission distances of 20 km within an urban environment and up to 35 km line of sight.

## 1. Introduction

The wastewater collection network (WCN) is a system which is rarely thought about, and even less frequently observed. In New South Wales, on the south east coast of Australia, the local water and wastewater management authority is Shoalhaven Water. The authority has over 4660 square kilometres of network coverage; over 18 million litres of wastewater is collected daily, and then distributed through the 1100 kilometres of gravity and pressure sewer mains for treatment. To put this into perspective, this equates to over seven Olympic-sized swimming pools of wastewater, daily. Effluent from homes travels via gravity pipe networks and collects for treatment. To service this system, Shoalhaven Water utilises a sewer pumping station (SPS) network of over 230 individual SPSs, transferring wastewater to one of thirteen sewer treatment plants (STPs). In between each SPS, wastewater is transferred through a combination of both rising mains and gravity mains, onto the next SPS and then lastly to the STP for treatment (Figure 1). A gravity main relies on gravity for flow while a pump is required for the pressurised rising main [1].

Currently, management of most sewer networks is more reactive than proactive, with response to blockages and potential sewer overflows in gravity systems only undertaken after these events have occurred. Every wastewater overflow event must be reported to the Environmental Protection Authority (EPA), with contamination of sensitive areas leading to potentially large monetary penalties [2]. Currently, overflow events at a SPS are recorded and alarmed through Shoalhaven Water’s Supervisory Control and Data Acquisition (SCADA) system. However, complications between each SPS, in the rising and gravity mains and their associated manholes are only brought to attention after these events have been reported, quite often by the public in either their private dwellings or nearby land. Some system intelligence can be implemented on the current SCADA system end to alert Shoalhaven Water engineers and managers of out of the ordinary flows between SPSs, however this still does not take into account the unknown condition and potential problems of the, in some cases, tens of kilometres worth of mains in between [3,4].

To proactively inspect these systems is rather costly, with estimates of labour, time, equipment and vehicles; sewer inspections cost on average between $6 and $15 per metre of pipe. Hence, to inspect the entire Shoalhaven sewer network would cost between $6.6 and $16.5 million, without taking into account accessibility difficulties and the logistics of all sections of the network on private land. These price estimates are compounded by the length of time it would take to inspect. The conditions of the initial sections of pipe to be inspected would significantly deteriorate by the time the final sections of main were complete [5,6].

Historical methods of directly measuring flow are conducted using flow meters that are highly invasive, requiring pipes to be excavated, cut and flow meters placed in series with the existing sewer network [7]. This method carries significant costs and inconveniences for both water utilities and residents. Furthermore, flow meters are only effective in the pressurised rising mains, where liquid is pumped out of the SPS. There is currently no feasible solution to comprehensively monitor the flow of wastewater within the gravity main networks [8]. A new process of WCN system management is required.

Following the initial research published in [9], this paper describes the experience of developing and installing two major projects that are intrinsically tied together:(1)A portable radar device that utilises the both Bluetooth WPAN and 3G WAN to measure flows at sewer pumping stations. In the paper we call this the Sewer Pumpstation Universal Drawdown Device (SPUDD)(2)An integrated radar device that interfaces to LoRaWAN network that continuously measures the state of the WCN and is permanently installed in selected manholes.

The importance of these studies lie in that they offer novel solutions to a number of the common difficulties shared by water utilities around the world:(1)A low-cost method to monitor the wastewater levels within SPS collection wells and, indirectly, accurately estimate the corresponding pump flows.(2)A real-world case study on the practical implementation of a LoRaWAN network to establish communications from remote sensors spread throughout a WCN, maintaining continuous monitoring.

In Section 5, we present a detailed review of similar systems for measuring flow rate. These include development of a two-dimensional (2D) sampling device [10], a computer vision-based system [11], a self-calibrating derived flow system [12], both mathematical and hydraulic modelling methods [13,14] and a real-time performance assessment system [15]. But the system that we present here is more practical, scalable and has a much lower cost that what is presented in the art.

Section 2 describes the methodology for this project, including reasoning and selections of device hardware. Section 3 includes implementation, results and analysis of all field-testing conducted. Section 4 provides a discussion of these implementations, while Section 5, Section 6, Section 7 and Section 8 include a review of current research, a background on LoRaWAN technology, conclusions and future works, respectively.

## 2. Methodology

In this section we describe the methodology for selecting and integrating the radar device which is used for both implementations.

### 2.1. Selection of Measurement Device

Figure 2 shows a typical manhole setup where the measurement device will be required to be located.

There are three main types of electronic level measurement devices, consisting of pressure transducer, ultrasonic and radar sensors. In order to select the best testing device for the required purpose, all were researched and evaluated.

#### 2.1.1. Pressure Transducer

The pressure transducer (which requires full contact with the effluent) was omitted due to its requirement to be constantly submerged which leaves it prone to collecting unwanted debris. In addition, the turbulent flow of liquids with suspended solids, result in fluctuating readings.

#### 2.1.2. Ultrasonic

Ultrasonic sound waves are used to send a signal down to the surface of the flow at the bottom of the manhole. The sound waves travel at a speed of approximately 343 m/s in air, which bounce back to be detected by the device. The main drawback with ultrasonic sensors is that false-positives can often occur in humid environments with steam, mist and water condensation. Similarly, where soft objects such as foam are present, the sound energy is absorbed instead of being reflected and the recorded return signal is minimal. Further, ambient temperatures greatly affect the speed of sound, *c*. For example, in air, *c* = 343 m/s at 20 °C, however *c* = 322, 352 m/s at −15 °C and 35 °C, respectively. Thus, in most industrial applications additional temperature compensation sensors are required for ultrasonic measurements.

#### 2.1.3. Radar

Radar devices utilise electromagnetic waves that travel at the speed of light, 3 × 10^8^ m/s, for level measurement. Radar devices operate similar to that of the ultrasonic sensor, sending out a direct microwave signal from the device which, as shown in Figure 3, is reflected from the water surface and received back by the device [16]. The delay of the signal is proportional to the depth of the liquid and the signal is converted into a very accurate measure of the liquid level.

Advantages of the non-invasive radar include that it is lightweight and easy to operate. The high frequency microwave signal is not affected by conditions such as humidity and mist, eliminating many of the false-positive readings returned by ultrasonic devices. The disadvantages include generation of false positives if the device is set up incorrectly and is more expensive than the alternatives.

We provide a summary of the radar operating principal in Section 6 (Background). Based on the above, a radar was selected for non-contact measurement, and Figure 4 shows a typical manhole setup with a radar sensor attached at the top.

## 3. Implementation and Results

Two radar sensor platforms were designed for this project as mentioned earlier; A sewer pumpstation universal drawdown device (SPUDD) and a LoRaWAN IoT-network enabled device. In addition to these, a third standalone device was built to perform signal strength measurements (radio channel ranging survey). In this section, we describe both device platforms, as well as their respective results. A general flowchart of this section is presented in Figure 5 below.

### 3.1. Sewer Pumpstation Universal Drawdown Device (SPUDD)

SPSs have historically had the flow rate measured by a method known as a drawdown test. This test involves using a tape measure to record the change of wastewater level within a SPS well, while a pump is running. This test is conducted over an initial period of 30 seconds, which is then used to indirectly calculate the outflow of the pumps. Inconsistencies in results have occurred due to measuring inaccuracies in both recorded times and measured medium heights, and also from both variable speed drives (VSDs) and soft start/stop operating methods. This test was deemed to be inaccurate as the amount of human error involved was high, and any use of VSDs directly skewed results. What follows is our unique solution that we developed to overcome the aforementioned problems.

#### 3.1.1. SPUDD Design

Summarised below is the field implementation design of the drawdown device.

(a) *Field Implementation Design*

The radar device already comes with a top threaded chamber, shown in Figure 6, that uses an LCD screen to show measurements. We have completely added a new second threaded chamber to the radar and an extended plastic housing has been developed, large enough to contain all relevant prototype microcontroller (MCU) and printed circuit boards (PCBs). This is also shown in Figure 6 below. We designed the housing with IP68 rating in mind (for waterproofing). The deep thread, rubber O-rings and triple screw locking mechanism preserve the devices original IP68 rating.

The grey screw cap on the end, shown more clearly in Figure 7 below contains a clear glass window to enable a view of the prototype boards, battery and real-time observation of status LEDs; including a flashing red to indicate the MCU searching for a Bluetooth device to pair, and a steady blue to indicate pairing to a Bluetooth device. This sealed, screw cap also provides easy access to the hardware modules contained inside. This design also allows for the ability to quickly connect a USB cable to the MCU for both charging of the battery and program modifications, as well as the capability to extract the microSD card from the Adalogger FeatherWing.

Figure 8 below shows the current prototype bracket, comprised of aluminium plate, one bar is fixed whilst the other is adjustable via a ring screw to allow mounting on top of various sized SPS well safety grates.

(b) *Testing Design*

The portable drawdown device on-site setup and testing is shown in Figure 9 below. As also shown in Figure 8 above, the drawdown device is mounted firmly within the SPS well safety grates. It is important to note that the device must not be mounted directly above a pump as it may look like below, but in the space in-between pumps to avoid false measurements returning from the top of the pump as the well effluent draws down towards the bottom of the well, exposing the submerged pump.

The experimental procedure included systematic testing of:(1)A standard Direct-Online (DOL) pump SPS,(2)A SPS with soft-starters to control the power ramp-up/down, and(3)A SPS with VSDs to control the power ramp-up/down of each pump.

At the outlined above SPSs, sixteen (16) tests were conducted on each individual pump to allow statistical confidence in results, utilising both the portable radar device and the common practice manual drawdown test with tape measure. Results were compared against one another, as well as against the quoted performance curves of the respective model of pump and its designed duty point.

(c) *Mobile Application*

In addition to the MCU programming, a simple Android Mobile/Tablet app has been developed to act as the user-interface for the radar drawdown device. The current mobile application chosen to interface with the radar device is pfodApp. The pfod acronym standing for ‘Protocol for Operations Discovery’, is quoted as allowing “Control Arduino and other microprocessor devices from your Android mobile—NO Android Programming required” [17]. This app acts as a data sink, and decoder of messages sent via Bluetooth from the MCU. The pfod protocol is a light weight protocol for simple device control and measurement. Each pfod message is specified to be less than or equal to 255 bytes in length, and is used to specify both objects, formatting and values for display on the pfodApp screen. The mobile pfodApp can be thought of as a micro-browser, displaying the micro-pages sent from the MCU.

Whilst the pfodDesigner app can be utilised to create your desired project controller app, and then generate the required Arduino code for programming; once the pfod protocol was learned it was found to be more effective to write the relevant pfodApp and Arduino code directly, interfacing with the required prototype devices. Following the flow chart shown in Figure 10a, Figure 10b contains the relevant designed app screens captured in a live, on-site test; from SPS Scheme selection, to specific SPS then pump selection, to the real-time measurement and graphical overviews.

#### 3.1.2. Results

Summarised below are the achieved results of all three test-cases.

(a) *SPS—Site A*

Site A comprises of one pump with flush valve (Pump 1) and one pump DOL (Pump 2). The flush valve fitted to Pump 1 is utilised to clean the base of the well prior to pump activation. When the Pump 1 Run contact is energised, for the initial 15 seconds the flush valve is activated prior to the pump running. This action stirs up the contents within the SPS well, allowing any settled solids to be flushed through the pump once running and not build up on the base of the SPS well.

Figure 11 below shows the time-series radar plots of outflow measurements. Samples were derived from both the current manual drawdown method as well as the prototype drawdown device. Manual tests here were conducted over a 60 second period.

These sample measurements were then normalised to a Gaussian distribution and plotted to clearly indicate variability between results. Figure 12 below shows these Gaussian functions. One can see that while both pumps manual drawdown tests are lower than that of the prototype radar device, the variability between Pump 1’s samples is significantly less.

This can be explained by the initial 15 second zero-outflow period whilst the flush valve is activated. This transient time significantly affects the manual drawdown test results as for one quarter of the flow measurement period, zero outflow is produced. Pump 2’s results however, do show a closer recorded outflow to the radar device as the DOL pump is producing outflow for the full sample period, though variability is much higher due to human error of recorded values. One can see that the mean recorded outflow rate for both pumps 1 and 2 at Site A by means of radar drawdown test are approximately 24 L/s, which corroborate the designed duty point for the pumps installed at this site, while both manual results are significantly less than this duty point.

(b) *SPS—Site B*

Site B is one of the larger SPS within Shoalhaven Water’s sewer network. It is the main pump station for one of the larger catchments, with the STP directly downstream from this site. As such, the required flow rate for this site is significantly higher than the previous SPS tested. Due to this demand, power ratings for the installed pumps at Site B are significantly higher, thus VSDs are installed for both pumps to not only limit the starting current with a 30 second ramp-up period, but to also limit the standard operating frequency to 40 Hz. This lower set frequency currently limits the outflow rate to less than the pumps typical designed flow rate, to allow for population and hence inflow growth over the design life of the pumps.

This SPS also has a magnetic flowmeter (Magflow) installed on the outgoing rising main. This station was selected specifically for this property. As such, we can use this flowmeter’s outflow results for further comparison against both manual and portable radar drawdown device results. Tests here were conducted over a 120 second period to cater for the 30 second VSD ramp-up/down periods. Figure 13 below shows the time-series radar plots of outflow measurements. Samples were derived from both the current manual drawdown method as well as the prototype drawdown device.

These sample measurements were then normalised to a Gaussian distribution and plotted to clearly indicate variability between results. Figure 14 below shows these Gaussian functions. One can see that both pumps manual drawdown tests are much lower than that of the prototype radar device, with significantly larger variability between sample results.

For the portable drawdown device however, both pumps recorded mean outflow rates measured at approximately 103 L/s. This outflow rate was also confirmed to be within 1 L/s of the recorded peak outflow rate of the Magflow at 102 L/s.

(c) *SPS—Site C*

Site C is classed as medium-sized within Shoalhaven Waters SPS network; being too large for DOL pump operation, yet too small for VSDs to be a cost-effective solution. As such, this site employs a soft-start/stop mechanism to limit the starting current of both pumps from the normal inrush current of 7–10 times, to less than 1.5 times that of the rated operating current. This mechanism is beneficial to electrical systems with limited current capacity; with Site C located in a combined residential and industrial setting, Shoalhaven Water cannot legally draw 7–10 times the rated current from the power grid of this site.

Soft starters also reduce the mechanical shock associated with motor start up; as the required torque to begin motor rotation from stasis is substantial, it provides a more gradual ramp up to full speed. Soft starters in this sense achieve the same function of a VSD ramp up, however in a much smaller form factor. Where Site B is comprised of a large building containing all electrical supplies, VSDs, Magflow and Human-Machine Interface (HMI) devices; Site C is a standalone pump control panel on the side of the main highway.

The soft-stop mechanism of the Soft starter is utilised at this site to provide a gradual ramp down to stop. This is to eliminate hydraulic shock, or ‘water hammer’ occurring in this particular wastewater network. This problem occurs when a high-flow pump stops running suddenly, causing a pressure wave to propagate within the pipe which often leads to a vibration or hammering noise within the network. Less frequently, however over an extended period of time, this phenomenon may even cause pipe within the network to collapse. This soft-start/stop mechanism provides a gentle ramp up in flow upon pump start, similar to that of a VSD ramp up, however creates more turbulent flow during the pump stopping period where after the ‘Pump Run’ contact is de-energised there is a secondary current peak at approximately 1.5 times the rated current. During this time additional wastewater is pumped through the well, long after the pump has been called to stop.

Figure 15 below show the time-series radar plots of outflow measurements. Samples were derived from both the current manual drawdown method as well as the prototype drawdown device. Manual tests here were conducted over a 60 second period, however Figure 15 shows the approximate 30 second soft-stop mechanism and associated secondary transient flow peaks.

Again, these sample measurements were then normalised to a Gaussian distribution and plotted to clearly indicate variability between results. Figure 16 below shows these Gaussian functions. One can see that both pumps manual drawdown tests are much lower than that of the prototype radar device, though the variability between sample results of both radar and manual methods are quite high.

This can be explained by the turbulent flows caused during transient stop periods, however, further investigation must be undertaken as to potential methods to eliminate the variability of the portable drawdown device measurements at sites equipped with soft-start/soft-stop mechanisms. One proposed method is to eliminate the secondary transient flow peaks from benchmarking to improve mean calculations. 

#### 3.1.3. Effect of Radar Accuracy on SPS Well Diameters

It is important to note here the effect of radar accuracy on flow measurements. Even with an accuracy of ±2 mm, this accuracy can cause a significant flow error for larger diameter SPS wells. As shown in Table 1 below, Site C’s well diameter of 2.54 m can cause up to an approximate 10 L/s flow error per second of level samples.

An exponential weighted moving average sample smoothing is employed to minimise these errors.

### 3.2. LoRaWAN Radar Network Deployment

A LoRaWAN base station (gateway) was installed within Shoalhaven City Council’s communication tower located on top of Cambewarra Mountain (Figure 17); a significant infrastructure asset at 650 m above sea level, this communications tower overlooks most of the Shoalhaven region.

With the gateway mounted within one of the site’s communications racks, the 915 MHz fiberglass omnidirectional antenna was installed on the transmit edge of the tower (Figure 18) to maximise the 3 dBd vertical beam width over the region.

An initial device was designed to first conduct the LoRaWAN signal strength measurements (radio channel ranging survey) within the Shoalhaven region. This consisted of a device constructed with a MCU and Multitech mDot LoRaWAN radio node [18].

A program was designed to send packets at five-second intervals, which included both Global Positioning System (GPS) location and an incrementing counter from end-device to gateway. The gateway measures received packet signal characteristics Received Signal Strength Indication (RSSI) and Signal-to-noise Ratio (SNR), attached as metadata, before forwarding the packet to the backend server. This allowed a comprehensive heat map of signal characteristics to be produced for the Shoalhaven region; showing maximum transmit distances localised to the region’s geographic area, as well as the corresponding signal statistics.

#### 3.2.1. Sewer Monitoring Design

Based on results achieved above in Section 3.1.2, the in-field application of these sensors will present a relative stable level, and corresponding flow, in average dry weather flows (ADWF), while the most beneficial data will be captured in rain events such as peak wet weather flows (PWWF). It is in the hours leading up to, during, and days following these events when sewer network trends are most vital to the development of the proposed non-invasive hydraulic model. A flow chart of the prototype device functionality is presented below in Figure 19a, as well as the intial vero-soldered device in Figure 19b.

#### 3.2.2. Results

A heat map of this initial radio channel ranging survey has been produced for the Shoalhaven region. Additionally, a prototype end device utilising the LoRaWAN protocol has been installed within Shoalhaven Water’s WCN to effectively monitor manhole SPS levels and flows.

(a) *LoRaWAN Network Capabilities*

Further analysis was performed on results collected from the Shoalhaven region network survey, and data received from the radar sensor node in a manhole upstream to the previously tested SPS Site B.

The radio channel ranging device design above allowed for a comprehensive heat map of signal characteristics to be produced for the Shoalhaven region; showing maximum transmit distances localised to the region’s geographic area, as well as the corresponding signal statistics. The incrementing counter sent from the MCU allowed a calculation of throughput over expanding radii.

**Throughput:**

Of a total 3100 packets sent, 2911 were successfully received by the gateway. Separating sampled data into expanding regions from the gateway however, demonstrates the effect of both distance, land topography and obstructions on signal throughput.

Table 2 below shows that as distance from the gateway is increased, the mean signal-to-noise ratio (SNR), received signal strength indicator (RSSI) and throughput all decrease, as expected.

Based on the above results, a heat map of this survey has been produced for the Shoalhaven region. Shown in Figure 20 below on the it can be seen that the majority of the Berry, Bomaderry, Nowra and South Nowra town centres are effectively serviced by this single gateway, with its reach extending as far as Gerringong to the North, Culburra to the East and Vincentia to the South. The Shoalhaven Heads township however is blocked from communications due to both Coolangatta Mountain and Moeyan Hill. Callala Bay and surroundings are blocked by the dense vegetation of the Currambene State Forest and Jervis Bay National Park.

To service these areas and avoid obstructions, an additional gateway is required on an alternative radio tower within range.

(b) *Radar Sensor Node*

Utilising the Cayenne LPP encoding scheme, the developed radar sensor device installed is capable of measuring the instantaneous well fill levels and calculating the corresponding derived flow rates. Both the well fill level and derived flow rate are transmitted via the LoRaWAN network and uploaded to the cloud-based platform; Cayenne IoT Project Builder Web Interface.

Currently transmitting derived values after a level deviation of 5%, the free Cayenne IoT Project dashboard provides not only an effective web-based tool to monitor any LoRaWAN devices deployed out in the field, but also offers a mobile app with close to the same functionality. On overview of the dashboard is shown in Figure 21 below.

Figure 22 below displays both the SNR and RSSI for each packet received by the Cambewarra gateway over a full-day period. It is shown that most packets arrive close to the mean SNR and RSSI of 3.42 dB and −100.65 dBm, respectively.

The daily signal statistics shown in Figure 23 below demonstrate that while all packets received were within acceptable range, there were a number of packets received that were well below the noise floor. All outliers for RSSI were, in fact, of clearer reception.

## 4. Discussion

Firstly, a novel portable radar device has been developed to effectively improve Shoalhaven Water’s WCN management procedures. A prototype device comprising of an 80 GHz short range radar sensor a low power, microcontroller and Bluetooth enabled device has been paired with a mobile application to offer an innovative pump testing method.

This paper offers detailed results and analysis of testing the prototype solution at specifically selected sites within Shoalhaven Water’s SPS network. This data yielded that this device is a much more accurate solution to the current manual drawdown method in all three initially proposed cases:(1)A standard DOL pump SPS,(2)A SPS with soft-starters to control the power ramp-up/down, and(3)A SPS with VSDs to control the power ramp-up/down of each pump.

Additionally, a single SPS pump fitted with a flush valve was tested, with supplementary data gathered in this pilot study propounding the view that this device is an innovative solution, currently rolled out into the field with relevant staff to undertake true Quality Assurance (QA) and Quality Control (QC) testing. SPUDD has proactively identified a number of performance issues within operational SPS before they have presented as severe problems. These include rising main blockages (chokes), damaged pump impellers and partially closed reflux valves.

A low-cost method to monitor the wastewater levels within SPS collection wells and, indirectly, accurately estimate the corresponding pump flows, this device offers a practical engineering solution to not only effectively monitor SPS pump performance, but also act as a portable initial solution to monitor the gravity sewer system for extended periods.

Further, a LoRaWAN network has been configured and deployed. Comprising of a single gateway and a number of prototype wireless sensor nodes, this paper has proven the effectiveness of LoRaWAN as a low power, wide area network for utilities in an urban environment.

Covering a large portion of the Shoalhaven region with a single base station, a comprehensive radio channel ranging survey was undertaken to measure both signal strength and reception characteristics. Furthermore, a prototype end device utilising these LoRa modulation and LoRaWAN protocol stack has been installed within Shoalhaven Water’s WCN to effectively monitor sewer levels and flows; offering a long range, low cost and low power alternative solution to that of a standalone sensor with 3/4G connectivity. This paper presents a real-world case study on the practical implementation of a LoRaWAN network to establish communications from remote sensors spread throughout a WCN, maintaining continuous monitoring. Whilst offering a number of utility-focused solutions in this paper, there is significant research being published around the world. With IoT and wireless sensor networks a current hot topic, a select sample of these works is reviewed in Section 5 below.

## 5. Related Works

The overarching focus of modern literature is the importance of efficient WCNs and their necessity for real-time monitoring and control; for not only supervisory requirements but also system performance analysis. This can be achieved through a number of means, including improvements to pump flow rate measurements and pump efficiencies, as well as the addition of accurate level measurement devices. There is also rapidly growing literature on long range, low throughput networks (LTN), indicating that these may offer novel solutions to the current boom of large scale wireless sensor networks.

### 5.1. Flow Rate Measurements and Pump Efficiency

In recent literature there have been many novel approaches to pump flow rate measurements and pump efficiency within large systems. These include development of a two-dimensional (2D) sampling device [13], a computer vision-based system [14], a self-calibrating derived flow system [15], both mathematical and hydraulic modelling methods [16,17] and a real-time performance assessment system [18].

Through development of a new 2D sampling device, named “Cerberes”, Larrarte [13] outlines a project undertaken to improve the accuracy of flow rate measurements in man-entry sewers. Cerberes utilises three acoustic Doppler velocimeters on a vertical jack to sample three simultaneous flow measurements along a vertical profile. Set up at an experimental site in France, the sewer line is rather large, 2.86 m in height, and 2 m in width at its widest section. A 130 m long section of the line was utilised to sample data from a number of sensors. Using the real-time data sampled from a combination of the abovementioned sensors, Larrarte was able to develop isovelocity maps of a vertical cross-section of the sewer main in both wet and dry-weather flows. Larrarte provides an excellent source of sewer main hydraulic information. While the sewer main used in [13] is combined sewer and stormwater; much larger than the mains within Shoalhaven Water’s network, Larrarte clearly displays the difference in isovelocity maps of the standard egg-shaped sewer and a section of non-standard shaped sewer.

A novel, vision-based system to monitor wastewater flow through image processing methods is presented by Nguyen et al. in [14]. Image-based water level algorithms have been used in partnership with the isovelocity maps found in [13] to compute the wastewater flow rate as a function of both the sewer main geometric profile and instantaneous height of water level. Nguyen et al. placed water level rulers on the walls of a number of sewer mains in Switzerland, capturing real-time video of the water levels through the use of Power over Ethernet (PoE) IP cameras. The camera’s recorded images are rectified through the homography matrix transformation of the two projective planes, allowing any point on the recorded sewer wall to be mapped into real-world coordinates and thereby perform metric measurements on the images. While the above in [13] used a more classical method of flow measurement, this developed system has proven to be effective in calculating wastewater flow rates, and their solution is impressive.

Another alternative method to monitor SPS pump inflow and outflow is discussed by Kallesoe and Knudsen in [15]. They state the difficulties with installing flowmeters at every SPS within a WCN, as well as the influence of operational wear on pumps affecting existing derived flow calculations. Unlike [13,14] above, this proposed flow estimation model has been tested within an operational SPS in Denmark. They have used a level measurement device to record SPS well levels in real-time, feeding this data into their flow calibration controller over a 20 hour, 121 pump start cycle period. Reference [15] found that the proposed self-calibrating estimation model is a cost-effective solution, and an alternative to installing costly flowmeters at SPS sites. The low cost allows water utilities to install a greater amount of devices within a network, without the need of recalibration of flowmeters.

Kallesoe and Knudsen do present an effective flow estimating solution for SPSs, however they have only focussed their testing and results on a direct on-line (DOL) pump station. The hydraulic models of SPS with soft start/soft stop or VSD operations are much more complicated than a binary start and stop model, as have been explored in this paper.

The mathematical model presented by Cui et al. in [16] allows optimisation of both location and control of SPS in a WCN. By refining controls of the strategically placed pump stations, Cui et al. propose modelling algorithms to reduce sewer overflows and provide substantial cost savings from both lowering system energy consumption and a decreased amount of pumping stations within a WCN. Cui et al. found that by optimising pump locations within a WCN, one can decrease the amount of overall pumps whilst still preventing sewer overflow in rain events. Consequently, by reducing the number of pumps within a network, significant energy and cost savings can be gained.

A hydraulic model which calculates the energy losses, by means of head pressure, in sewer manholes is proposed by Merlein in [17]. These results provide confirmatory evidence that by installing perforated cover plates on sewer manholes, head loss coefficients can be reduced which, in turn, can reduce the water levels inside manholes further upstream in wet weather events. Both Merlein’s hydraulic and mathematical models will become an excellent resource for future research, when mapping Shoalhaven Water’s selected sample sewer network.

An online, real-time performance assessment system which allows water utilities to monitor specific performance indicators within a WCN is offered by Kooij et al. in [18]. Utilising the real-time data and model simulation results, the problems associated with underperforming systems are highlighted. The commonality of post-incident data analysis as opposed to predictive monitoring are contrasted by proposed new technologies which can provide significant operational cost-savings. Kooij et al. have implemented an in-system monitoring scheme to monitor specific performance indicators. Integrating pump run, pressure, power and flow data from a SCADA system, and hydraulic model calculations from a pipeline software system, a new assessment tool was created. Whilst the integrated control system for energy cost savings had not been implemented, a numerical model of the forecasted savings was presented. By monitoring these specific four performance indicators within an integrated assessment tool, significant savings could be made in both operational costs, and maintenance.

### 5.2. Radar Technology

A more straight-forward approach to improved SPS well level measurements, and hence derived flow data is to upgrade the predominantly float-based systems with commercially available measurement devices such as radars.

A novel, low power Frequency Modulated Continuous Waveform (FMCW) radar system is offered by Brumbi in [19] for continuous level gauging in tanks. The system design and functionality is outlined for the first industry-ready two-wire level radar sensor. Two-wire sensors are preferred for industrial applications as they are deemed to be “intrinsically safe” for use in hazardous applications. The added benefit of two-wire instrumentation is that they are loop-powered, eliminating the need for an additional instrument power supply. This 4–20 mA current loop signal is fed to an analogue input of a Programmable Logic Controller (PLC), where further processing occurs for control and monitoring purposes. This low-current loop allows for the power supply to be some distance from the required sensor, enabling exclusion from hazardous environments. Although somewhat dated, Brumbi outlines and explains the benefits of the now popular industry standard two-wire instrumentation, correctly forecasting radar devices to be the “future standard in level measurements—usable for nearly all applications”.

The antenna impact on the gauging accuracy of radar sensor level measurements are evaluated in [20]. By means of a MATLAB radar system simulator, the physical behaviour of a real radar test range can be implemented via a virtual test bed. This initial MATLAB simulation allows the ability to optimise antenna design and compare the impacts of various designs to level accuracy in the initial development processes. They offer insight to the presence of an analytical antenna model, which will prove to be invaluable in the future development of radar antenna design, where user-definable accuracy levels within the sub-millimetre range are required.

Furthermore in [21] Arbrecht et al. investigate the implementation of various dielectric tube antenna designs for industrial radar level sensors. They outline the geometrical restrictions placed on antenna design for industrial applications; with maximum outer diameters limited by standard process flange diameters, and moreover their maximum lengthwise extension is limited to ensure the radar sensor remains non-contact from the measured medium.

On the contrary and following on from [19], Arbrecht et al. explain the requirements of the external antenna to be completely of chemically inert dielectric materials such as polytetrafluoroethylene (PTFE) or polypropylene (PP) for use in harsh industrial environments. These inert dielectric materials are in fact advantageous as they offer superb radio frequency (RF) properties, with low dielectric losses and a steady permittivity over a range of both frequencies and temperatures.

Arbrecht et al. offers another alternative for possible signal directivity improvements, which may be required to be explored after initial prototype testing of the proposed radar device in a sewer gravity main environment. Utilisation of a radar complete with a PTFE or PP dielectric horn waveguide suggests a smaller device form-factor, as well as a reduced “antenna ringing”.

### 5.3. Low-Throughput Networks

A number of wireless LTN technologies are examined by Margelis et al. in [22], discussing the suitability of networks like these for certain industrial process applications and findings presented in the first peer-reviewed paper of this nature. Margelis et al. define LTN’s as low data rate and high range networks with specific characteristics including; data transmissions in the range of tens of kilometres, power efficient hardware with target battery life of some years and communication protocols to further enhance battery life.

The three LTN technologies presented include currently-proprietary protocols by companies Sigfox, OnRamp Wireless and the LoRa Alliance. With a LTN one possible “back to base” data communication method being explored for future works, a comparison of their technology is presented in Table 3.

The latest innovation boom of smart cities and the drive for a mass interconnected world, coined the “Internet of Things” (IoT) is discussed by Filho et al. in [23]. Whilst most worldwide utilities are already utilising this in some way by the means of RF mesh technology through remote site monitoring and smart metering, Filho et al. compare the emerging Long Range Wide Area Network (LoRaWAN) technology to the RF mesh for use in IoT applications. It was found that LoRaWAN-capable devices provide a much lower cost and open-standards based solution. LoRaWAN is much simpler in terms of terminal access, where each node has its own direct access to a base station, or gateway, compared to the complex routing protocols required for RF mesh. Due to this, LoRaWAN has been recommended to utilities which are yet to employ RF mesh for smart metering. The major gap with this research is the lack of experimental or simulated design. Whilst the technical framework and comparisons between the technologies are highly valuable, some form of analysis to substantiate the claims could have been conducted.

One of the first to analyse LoRaWAN’s medium access control (MAC) protocol, Bankov et al. in [24] explore the protocol’s varied spreading factors and bit rates. Simulated through MATLAB, a LoRaWAN network with *N* nodes, connected to a single gateway and using 3 channels at varied transmission probabilities, they have studied the relationship between transmission reliability and node-to-gateway ratio. The study found that the packet-loss ratio remains very small for low-load networks (0.001), for example that of 100 nodes each transmitting once every 20 minutes. If node numbers increased to that say of 5000 nodes, each would only be allowed to transmit 2 packets per day to maintain this packet-loss ratio. Research dictates that as the load increases, LoRaWAN becomes significantly less effective for a high number of nodes. It is noted that to increase LoRaWANs efficiency in higher-load networks, one must increase the amount of gateways within the network, thus decreasing the node-to-gateway ratio.

Mikhayalov et al. in [25] have published one of the first papers investigating the network capacity and scalability of LoRaWAN technology. Using the LoRa channel attenuation model, they have derived the cell capacity of amount of nodes to a single gateway while also producing theoretical upload link transfer speeds for various ranges. This research has shown that the LoRaWAN network is highly scalable under low data rates. Like [23,24] above, all results are based on the European frequency regulations and the 868 MHz Industrial, Scientific and Medical (ISM) band. Further research must still be conducted into the Australian frequency regulations and the 915 MHz ISM band.

An experimental network within a building, comprising of a single gateway and 13 nodes was set up by Neumann et al. in [26]. Placing nodes in different rooms at distances varying from 0.5 to 60 metres, throughput, delays, RSSI, SNR, packet loss, packet error and power consumption of the individual nodes were measured and compared within the network. It was found that, set to a 1% duty cycle, LoRaWAN data rate 5 (125 kHz channel, 5.47 kbps, 242-byte maximum payload) to be the most effective configuration for time-sensitive applications. Neumann et al. established that LoRaWAN technology can be effective within an indoor environment for IoT applications, as long as gateways and nodes are strategically placed. While experiments were conducted with the use of a single gateway for all nodes, and noting the relatively small cost of gateways for practical implementation, one would expect that using multiple gateways within an indoor environment will decrease packet loss from nodes a further distance from the original gateway.

Building upon the research found in [26], Wixted et al. in [27] tested both purely LoRa wireless RF and LoRaWAN technologies to compare the performance and reliability of each in a well-developed environment. This work by Wixted et al. is the first published research using a central business district (CBD) of a city, in this case Glasgow (UK), as a test-bed for this technology. Conducting two separate tests of both purely LoRa wireless RF and LoRaWAN networks, it was found that LoRa wireless RF communication was successful within an approximate 2 km radius from the static RF transceiver, even amongst buildings, with a healthier RSSI from line-of-sight communication. Conversely, signal was lost after passing over a large hill. On the LoRaWAN network however, performance increased significantly with a radius of more than double that tested in LoRa RF and a connection rate of 95.5%, with another 2.5% with data received but no acknowledgement. The LoRaWAN node also still transmitted in problem areas which even GPS signal was lost. It was established that both city topography and building density affected signal propagation. This will need to be accounted for in this project’s testing procedures and gateway positioning.

As previously discussed, this relatively new technology does not have many test-cases for large-scale deployment as yet. Significant testing will be required to be undertaken to prove the effectiveness of this technology in a Water Utility network. Wixted et al. concluded that a possible use case for this technology is within environmental monitoring applications, and it is envisaged that this paper will be one of the first to publish a proven use case in this field.

## 6. Background

### 6.1. Radar Operating Principles

The two main operating principles used in radar level measurement consist of pulse and FMCW (broadband) radars. To calculate the distance of the medium away from the radar device, a “time of flight” method is applied. This method measures how long a transmitted electromagnetic (EM) signal takes to reflect off the medium, and to be received again by the device.

Since the EM signal travels at the speed of light, C, the medium range, D, is known to be twice the time delay, Δt, multiplied by the speed of light:2D = CΔt,(1)

From (1), the medium range can then be calculated by:D = (CΔt)/2,(2)

Both principles utilise the same method of measuring range, with only a varied means of measuring the time delay, Δt.

#### 6.1.1. Pulse Radar

In a pulse radar device, the time delay is directly measured by the delay of each repetitive transmitted pulse between transmitter and receiver. Figure 24 below shows the method of pulse radar measurement.

#### 6.1.2. FMCW Radar

An FMWC radar however, emits a continuous waveform whose frequency increases linearly during the measurement phase, known as a frequency sweep. Its frequency is modulated, generally in a saw tooth pattern to distinguish between signal peaks. Figure 25 below shows the general principle of a FMCW radar measurement technique.

At any given time, the known slope of the saw tooth pattern, m, is given by:m = Δf/Δt,(3)

Hence, from (3), the time delay can be calculated as:Δt = Δf/m,(4)

Furthermore, this frequency difference Δf is calculated using a Fast Fourier transform (FFT). The difference is directly proportional to the distance of the medium away from the radar device, and is calculated from the FFT frequency spectrum.

For these specific applications, a radar device with high performance accuracy, minimal beam angle and low power supply requirements is essential. Figure 26 below shows the effect of radar beam angle on the beam width diameter at varied medium distances.

The deepest sewer manholes within Shoalhaven Water’s network are 5 m in depth which, with a beam angle of 3°, translates to a beam width diameter of approximately 260 mm. Compared to beam angles of 10+°, significant sections of both the gravity main pipe edges as well as manhole walls will be reflected.

Whilst most gravity sewer mains are approximately 150 mm in diameter, at maximum medium distance more than half of the radar beam is still reflected off the actual medium rather than sewer main and manhole walls. Once sewer levels are above the 150 mm main, water begins to vertically fill the 1500 mm diameter manhole. The VEGA VEGAPULS 64 [16] radar has been chosen, as it has the lowest power requirements but also the highest signal frequency, hence resulting in the smallest beamwidth and most accurate measurement. The PULS 64 also offers built in false signal suppression where others do not.

### 6.2. LoRaWAN Background and Implementation

The section describes the LoRaWAN technology and how it was implemented.

#### 6.2.1. LoRaWAN Communications Protocol

The LoRa modulation techniques were investigated further, as well as both the LoRaWAN MAC, security and a specific free-to-use network backend.

(a) *LoRa Modulation*

LoRa is a recent physical layer modulation technology developed by Semtech Corporation [28]. Utilising a proprietary Chirp Spread Spectrum (CSS) technique, LoRa spreads a narrow-band signal over a wide bandwidth channel. From the resulting noisy characteristics, LoRa allows for bidirectional communication with the ability to detect severely attenuated LoRa packets up to 20 dB below the noise floor. Once a transmitter broadcasts a LoRa packet, a receiver utilises a Fractional-N Phase-Lock-Loop to lock onto the preamble and begin reception of the packet stream.

A combination of the CSS and 4/5 FEC coding rate provides a receiver sensitivity of up to −137 dBm, enabling a strong resilience to channel noise and interference.

LoRa Alliance specifications [29] outlines the provision of numerous spreading factors (SF7-SF12) supported by LoRa. These spreading factors (orthogonal codes) offer a trade-off between data/bit rates and transmit range. Table 4 below shows that the larger the spreading factor, the lower the bitrate and receive sensitivity, and the longer the time-on-air.

(b) *LoRaWAN MAC and Security*

The LoRaWAN specification utilises an unslotted ALOHA scheme at the MAC layer which, on top of the LoRa physical layer, allows for communication between many multiple devices at the same time, on different channels and/or spreading factors; offering robust communication in even busy ISM-band channels [30].

The packet encryption techniques, for both ‘network session’ and ‘application session’ allows end devices to broadcast to multiple gateways, and hop from base station/s to base station/s without extra signalling overheads.

Packets are encrypted end-to-end to AES-128 standards; using a 128-bit network session key; which is known only by the LoRa module on the end device and the network server, and a 128-bit application session key; which is known only by the LoRa module on the end device and the application server. The encrypted packet is sent over the air, with the intermediate LoRa gateways and corresponding network server only forwarding encrypted packets to the application server. This eliminates the ability for packet sniffers to read sensitive data over the air, as well as reducing the networks cyber-attack surface for bad actors.

(c) *The Things Network Backend*

The Things Network (TTN) is a “global, open, crowd-sourced Internet of Things data network” [31]. Utilising their backend for this project, it is this network server that is responsible for decoding the data packets forwarded from the gateway; distributing to the correct application server based on the application session key. As shown in Figure 27 below, packets are broadcast from the end devices, to corresponding gateways in range, forwarded to the designated network server and then further to the application server.

TTN’s network server is responsible for the de-duplication of packets if received by more than one gateway. This is whereby if more than one packet is identified with a header including the same device address, frame count and timestamp, only the packet with the highest SNR and RSSI is kept, while the duplicate packets are dropped. An example of the event data captured from each uplink by the TTN network server is shown in Figure 28 below.

TTN’s web interface allows useful monitoring of packets both sent and received by end devices and user-owned gateways, however to process, store, analyse and/or display this information, further communication between third-party application programming interface (API) sources is required to provide access to various dashboard-style web interfaces.

#### 6.2.2. Cayenne Packet Encoding

To translate the measurements obtained utilising technology from the portable radar device, a method to encode the data into a packet size of less than 11 bytes (LoRa Data Rate 2, Spreading Factor 10) was required. The Cayenne Low Power Payload (LPP) [32] provides a convenient and simple encoding technique to send data over LPWAN networks such as LoRaWAN. The Cayenne LPP is compliant with the spreading factor payload size restrictions, and allows the device to send multiple sensor data at one time. For this sensor two values are required to be monitored; both well level (metres) and flow rate (litres per second), and is encoded as follows:



8 bytes per packet = 2 ‘channels’ worth of data:



***Channel 1: Level (metres)***

1 Byte   = Channel 1 Identifier

= 0x01

1 Byte   = Channel 1 Type Identifier   = 0x02 **(Analog Input)**

2 Bytes = Channel 1 Data       = 0x00 to 0xFF **(Range)**

= −327.68 to 327.67 **m**



Data Resolution = 0.01 **(Per bit, Signed)**



***Channel 2: Flow (L/s)***

1 Byte   = Channel 2 Identifier 

= 0x02

1 Byte   = Channel 2 Type Identifier   = 0x02 **(Analog Input)**

2 Bytes = Channel 2 Data       = 0x00 to 0xFF **(Range)**

= −327.68 to 327.67 **L/s**



*Data Resolution = 0.01 **(Per bit, Signed)***


## 7. Conclusions

While most theory and research has focused on indoor test-cases and reception limitations, there are limited larger WAN capability surveys. As part of this paper our results have far exceeded the theoretical communications capabilities in a WAN setting, boasting above a 70% throughput rate still at radial distances of 20–30 km. Although the mean SNR at these distances was found to be −7.54 dB, the LoRa modulation techniques have proven the ability to detect severely attenuated LoRa packets up to 20 dB below the noise floor

From the literature reviewed above, and works laid out within this paper, a strategy has been formed for future works to implement a large, scalable wireless sensor LTN utilising commercially available radar devices for measurement. Utilising fixed radar level sensors in SPS, and continued improvement in SCADA data collection and prediction has allowed for the plotting of standard, expected diurnal inflows and outflows at critical wastewater sites. This gives the ability to alarm in real-time lower than (or greater than) expected flows at specific time-slices of a day, identifying whether there is a potential problem either upstream or downstream from the corresponding SPS site. The proposed IoT radar sensor network will greatly supplement the current capabilities of this diurnal management tool; providing significantly more WCN data sets to further improve proactive location identification of potential system problems.

The major aim of this is to provide a solution to eliminate dry-weather sewer overflows completely from not only Shoalhaven Water’s network, but to assist in providing a future framework for all water utilities to implement.

## 8. Future Works

Following on from these works, a continued partnership with the University of Wollongong is envisaged to develop a novel solution, ready for mass production, to improve the water industry’s wastewater flow management systems. Building upon the prototypes already developed, this project will comprise of an integrated, wireless, 80 GHz short range radar sensor network that will report, in real time, on gravity sewer main water levels system-wide. Installed within Shoalhaven Water’s sewer manholes, each device will capture more flow data from the currently unmonitored areas between each SPS. Strategically placing approximately five devices in the gravity sewer network around each of the 230 SPSs, will allow for an effective solution to proactively capture the state of the gravity sewer system and, over time, create a non-invasive hydraulic model of Shoalhaven Water’s entire sewer system.

If we take a SPS failure as an example. A failure in one station will cause the build-up of sewage in the immediate area but will also have carry on effects onto other SPSs. Currently, events have to be dealt with on a case-by-case basis. All of these stations have remote monitoring, however manholes do not and are only assessed by regular physical inspection. These further down the line effects are only understood by the staff who have monitored the system for a long time. If a problem exists in part A of the network, then we have X number of hours before part B will be affected. What is even less understood is when you combine a wet weather event (excessive rain in one part of the network), in addition to a random set of events such as main blockages/chokes, a SPS failure (due to a number of reasons), or long-term deterioration in a single link of the network. Currently none of this is modelled with any accuracy, if modelled at all.

A robust monitoring system that includes the latest research in radar technology, low power wireless networks and statistical analysis aims to reduce the effects of such spills and/or eliminate them completely. Better data collection will allow for better development of models that can be used in the building of future WCNs.

As such, it is envisaged this network will have the ability to support upwards of 1000 radar sensors to operate as a single, cohesive network. Once the hydraulic model based on historic and real time data is developed, tested and functionally proven, it is anticipated that an innovative intelligent system will be established to predict overflow events before they occur. Based on historical rainfall, flow and sewer level data a neural network model will be trained to predict overflow events before they occur. The network flows need to be monitored effectively to produce an accurate snapshot of real-time network conditions and improve the environmental impact of the ecosystems. Furthermore, monitoring this network will provide significant savings in both visual inspections, reactive emergency work and the associated costs of overflow events.

Feeding all the collected, derived and predicted data into a corporate Business Intelligence system will enable not only Shoalhaven Water, but utilities worldwide to better manage their networks with less emergency response to wastewater overflows and the corresponding complications associated with these events.

This will allow forward planning of augmentations to existing network infrastructure to made based on inform decisions, and thus reducing both capital and operational expenditures.

## Figures and Tables

**Figure 1 sensors-19-00254-f001:**
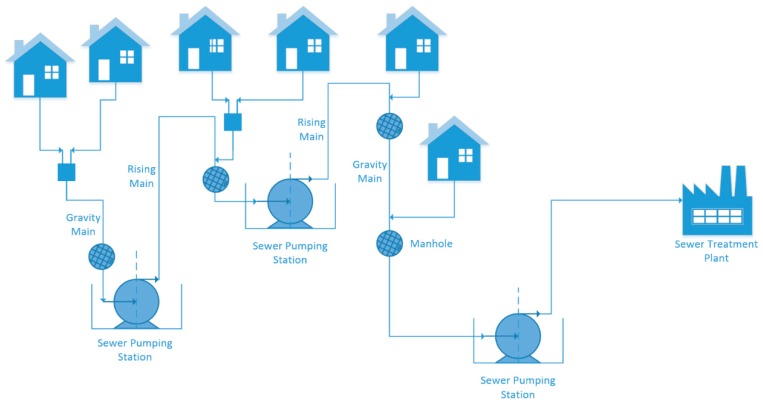
A typical sewer network.

**Figure 2 sensors-19-00254-f002:**
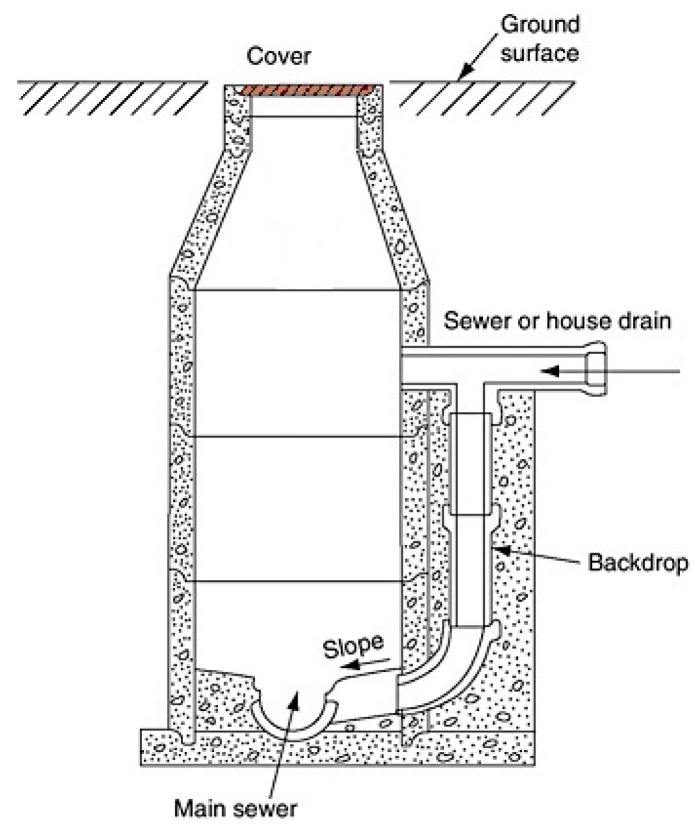
A typical manhole collection well.

**Figure 3 sensors-19-00254-f003:**
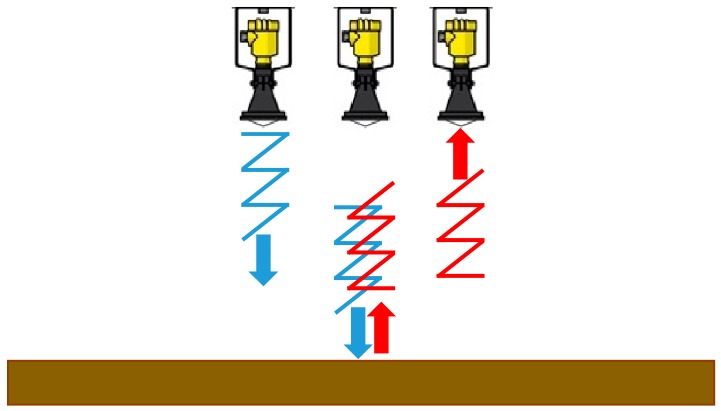
Function of a non-invasive radar.

**Figure 4 sensors-19-00254-f004:**
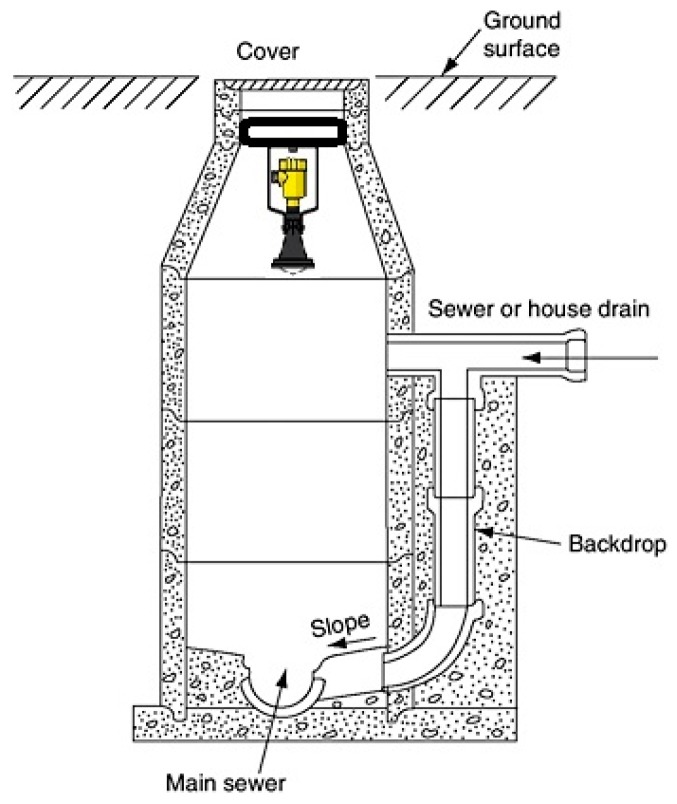
A typical manhole with radar sensor.

**Figure 5 sensors-19-00254-f005:**
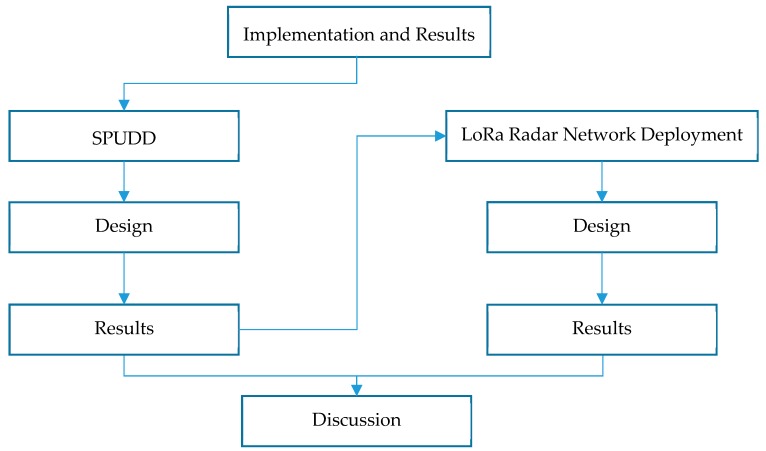
Implementation and Results flowchart.

**Figure 6 sensors-19-00254-f006:**
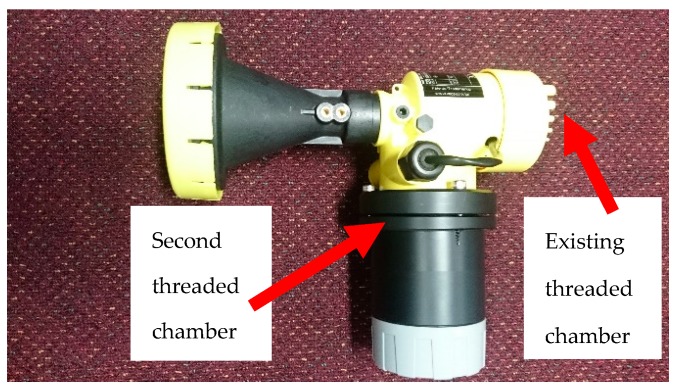
Portable radar chambers.

**Figure 7 sensors-19-00254-f007:**
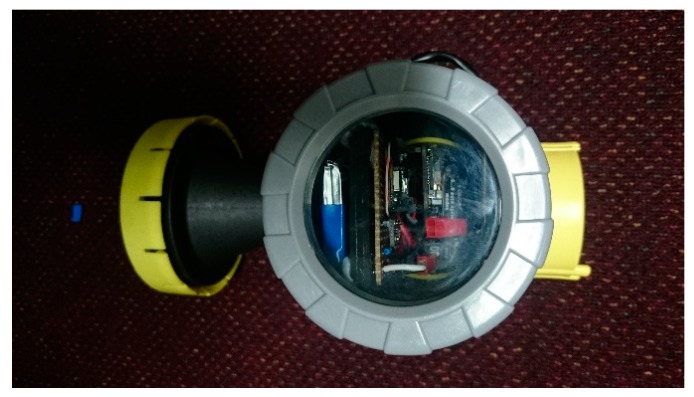
Chamber two glass window.

**Figure 8 sensors-19-00254-f008:**
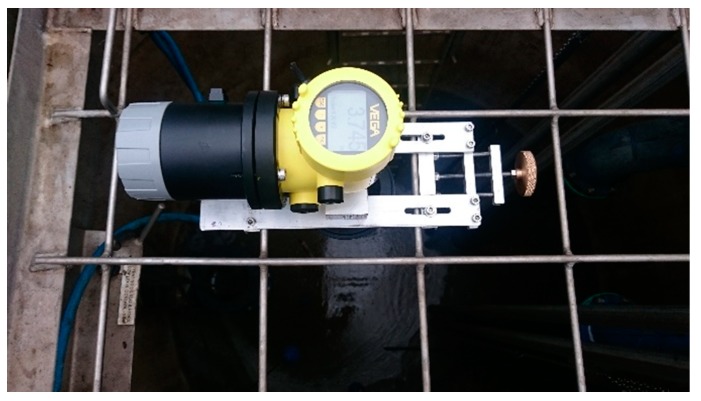
SPUDD adjustable mounting bracket.

**Figure 9 sensors-19-00254-f009:**
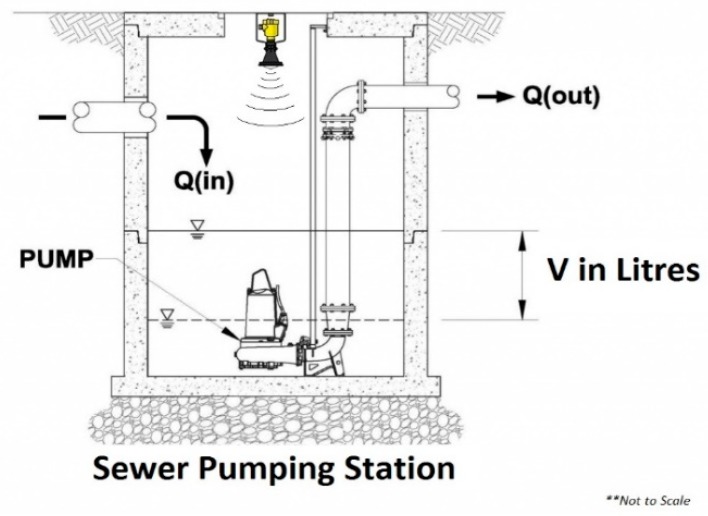
SPUDD on-site setup.

**Figure 10 sensors-19-00254-f010:**
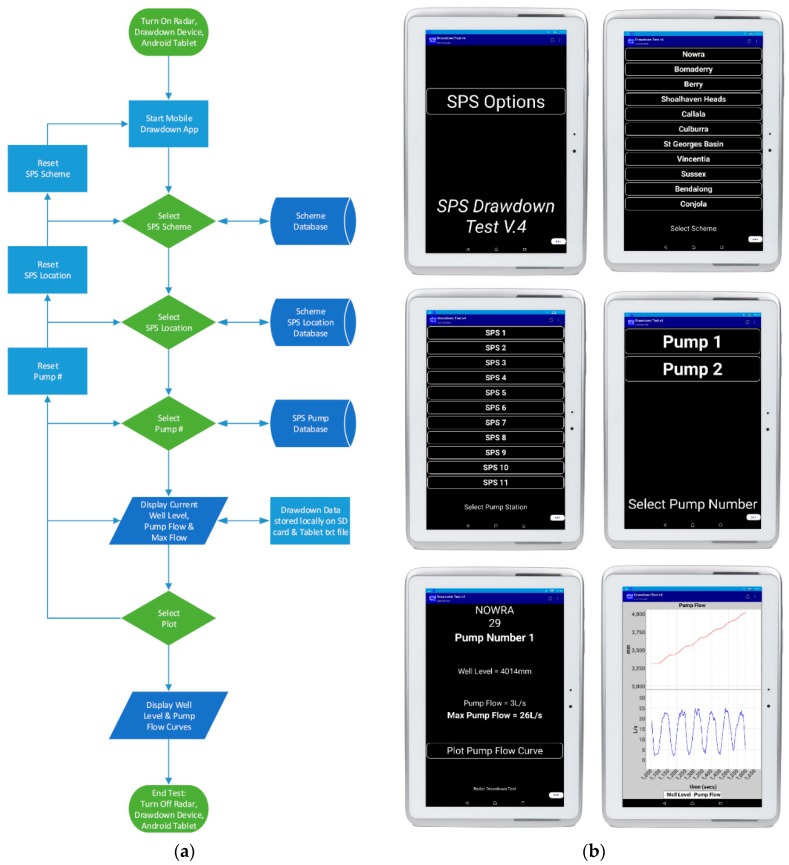
SPUDD (**a**) Operational flow chart, and (**b**) Mobile application screens.

**Figure 11 sensors-19-00254-f011:**
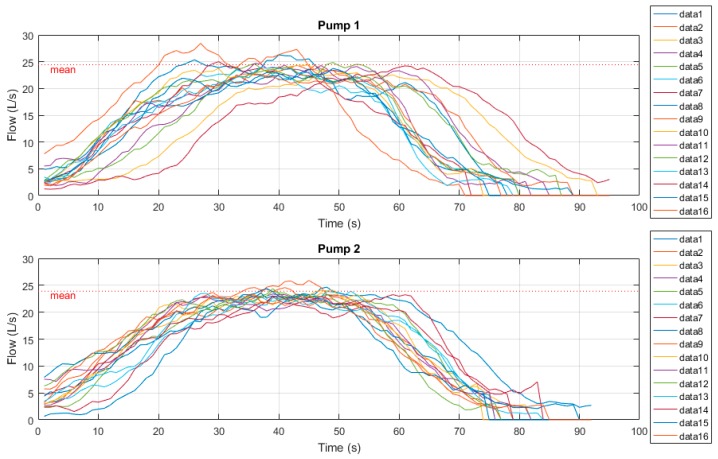
Site A time series radar plots.

**Figure 12 sensors-19-00254-f012:**
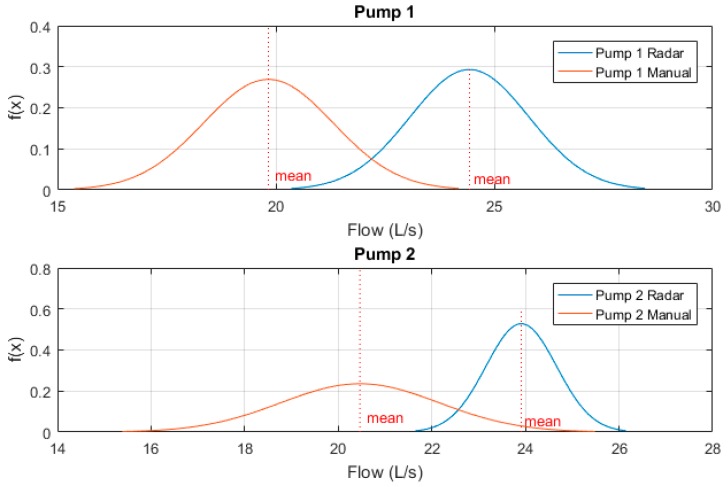
Site A normalised radar vs manual testing results.

**Figure 13 sensors-19-00254-f013:**
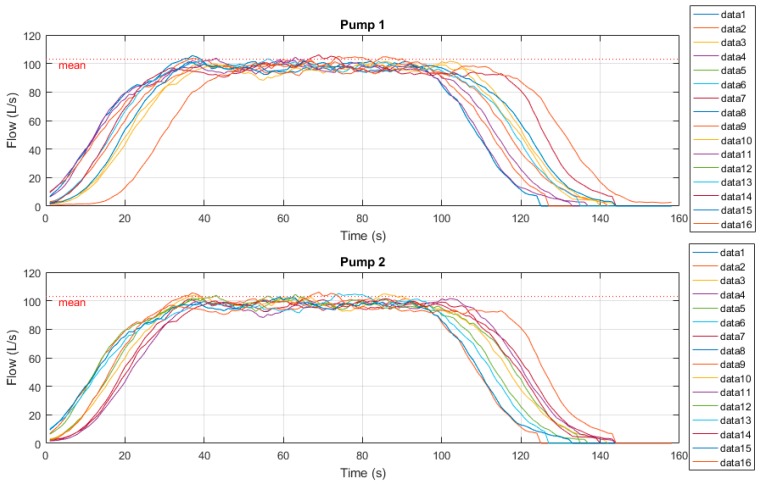
Site B time series radar plots.

**Figure 14 sensors-19-00254-f014:**
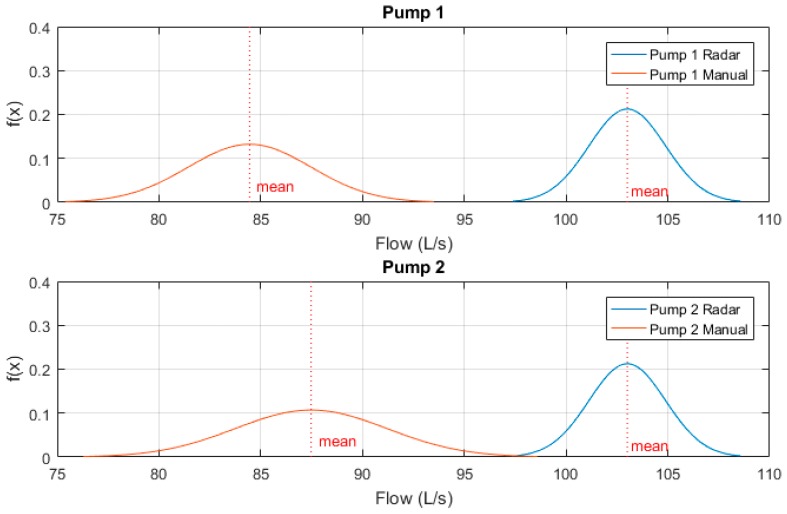
Site B normalised radar vs manual testing results.

**Figure 15 sensors-19-00254-f015:**
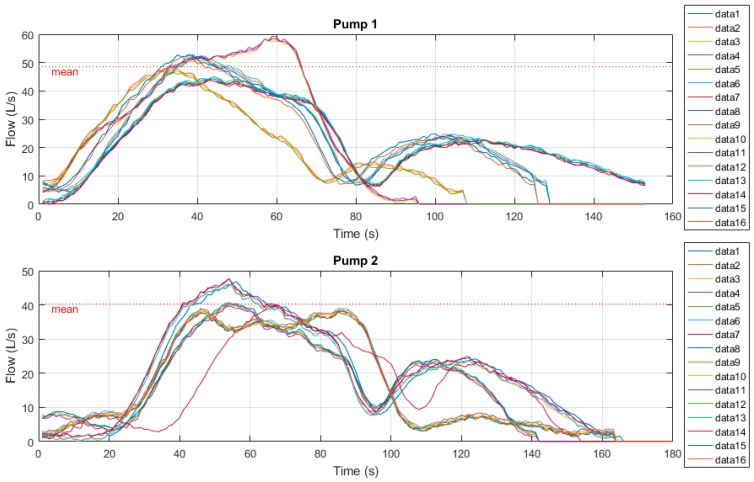
Site C time series radar plots.

**Figure 16 sensors-19-00254-f016:**
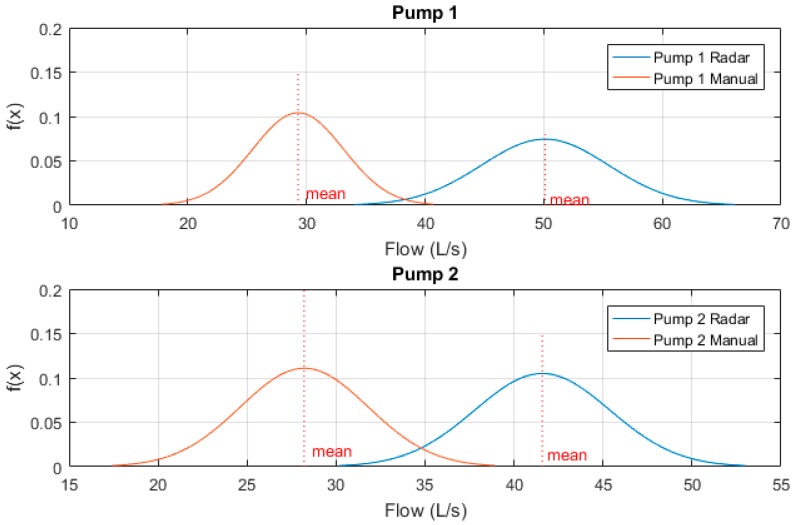
Site C normalised radar vs manual testing results.

**Figure 17 sensors-19-00254-f017:**
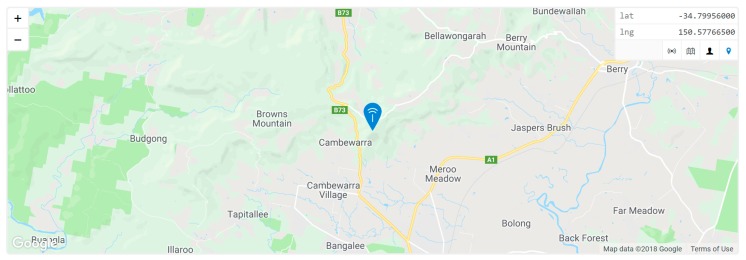
Cambewarra base station placement.

**Figure 18 sensors-19-00254-f018:**
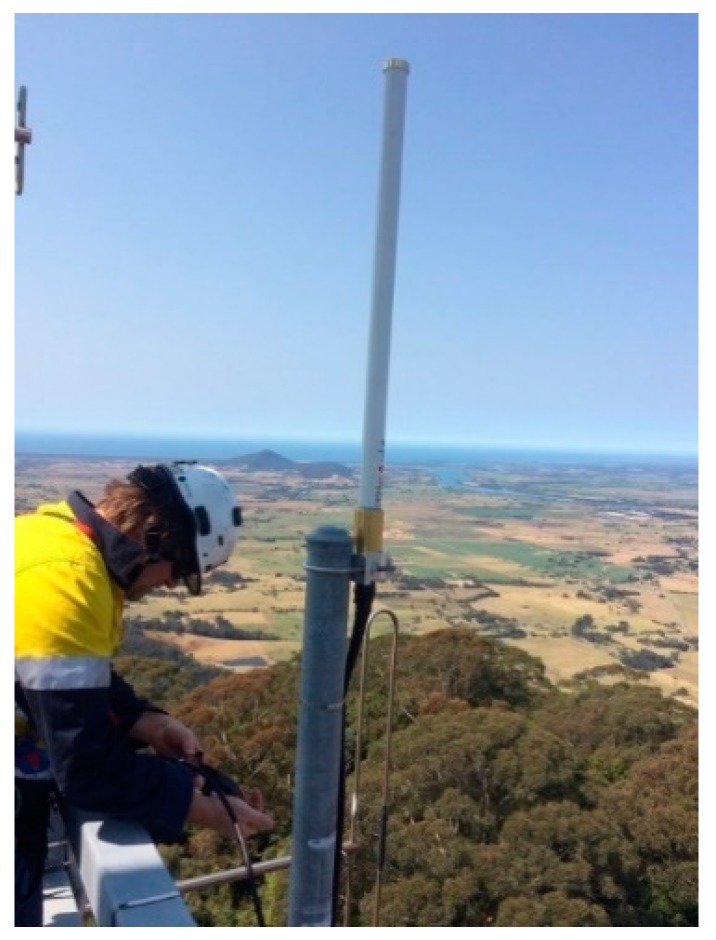
Omnidirectional antenna placement.

**Figure 19 sensors-19-00254-f019:**
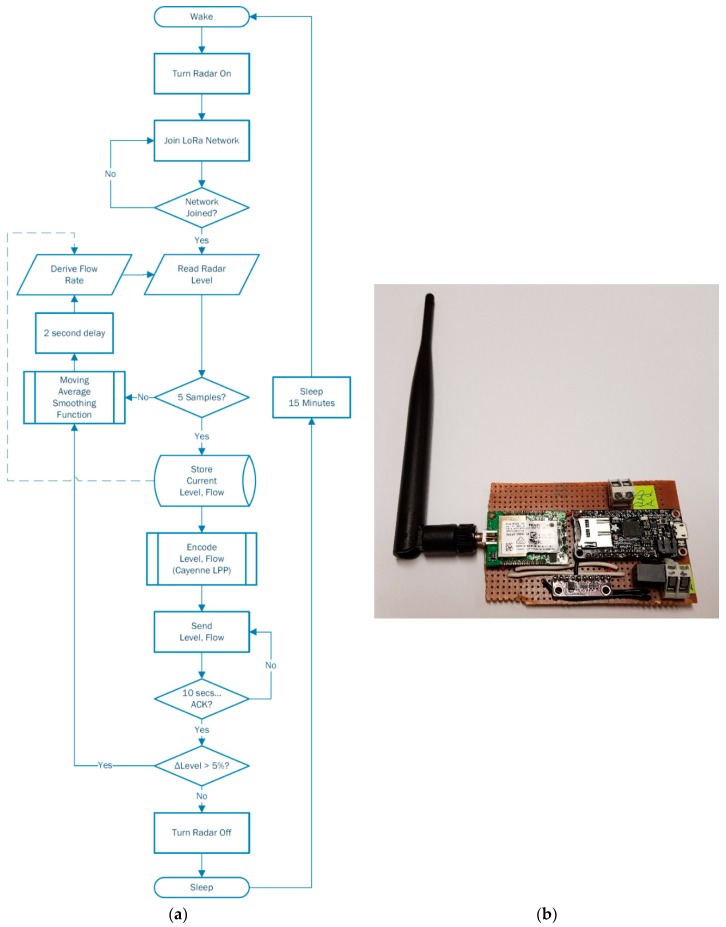
Prototype LoRa radar sensor device (**a**) Flow of operation, and (**b**) Interface board.

**Figure 20 sensors-19-00254-f020:**
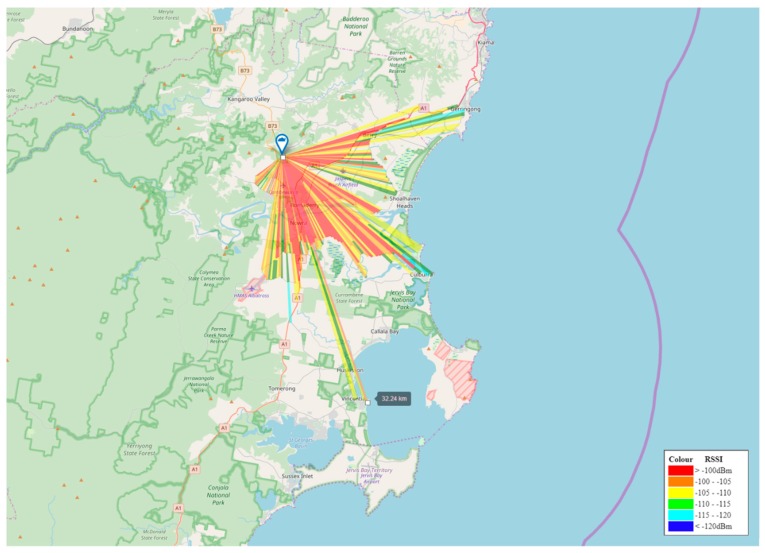
Shoalhaven region radio channel ranging survey.

**Figure 21 sensors-19-00254-f021:**
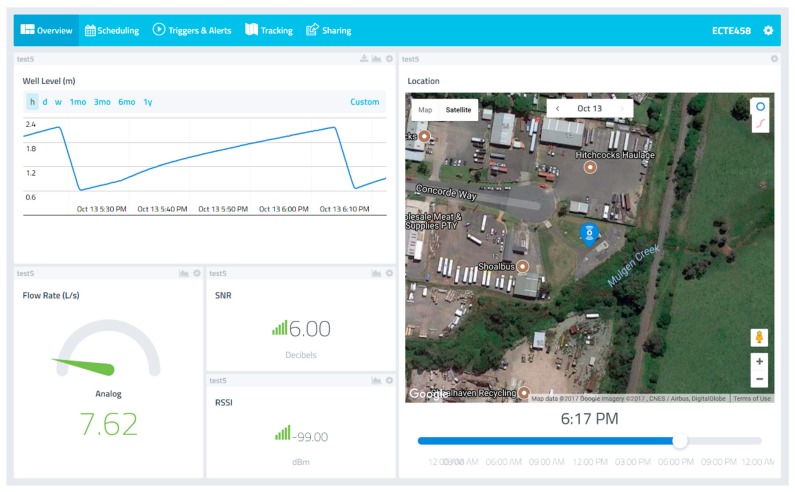
Cayenne IoT Project Builder.

**Figure 22 sensors-19-00254-f022:**
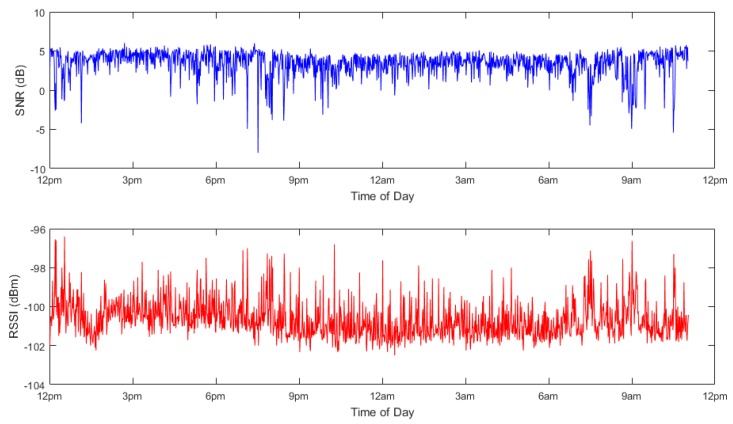
Daily signal characteristics—Site B.

**Figure 23 sensors-19-00254-f023:**
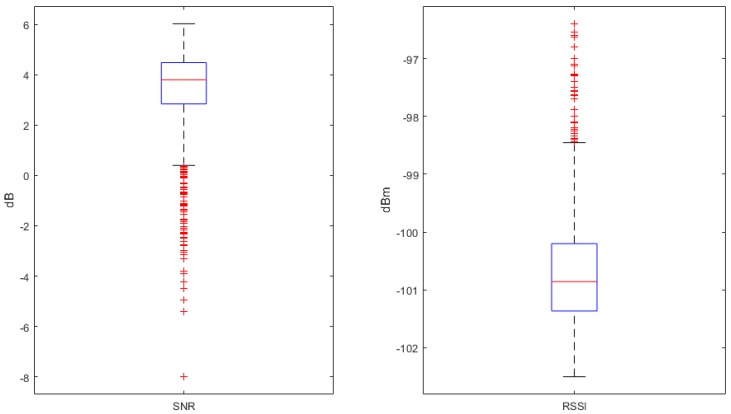
Daily signal statistics—Site B.

**Figure 24 sensors-19-00254-f024:**
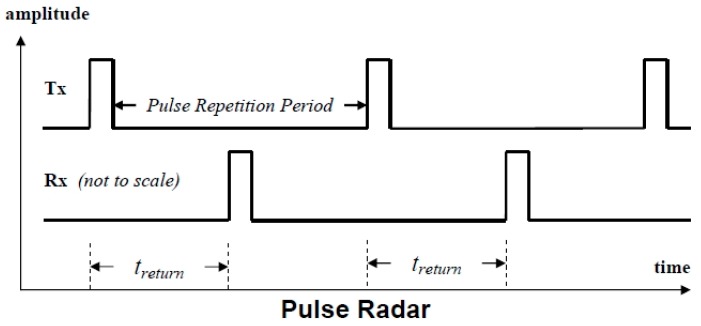
Pulse radar measurement principle.

**Figure 25 sensors-19-00254-f025:**
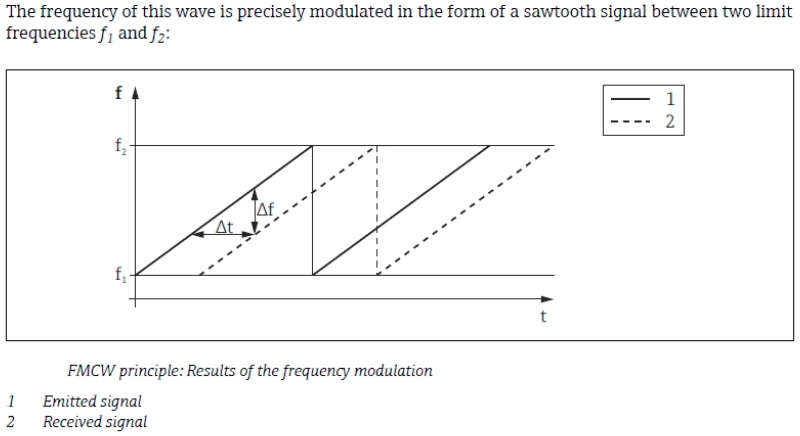
FMCW radar measurement principle.

**Figure 26 sensors-19-00254-f026:**
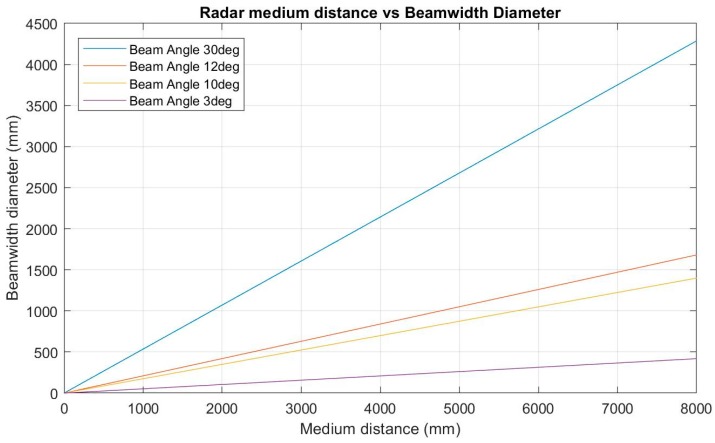
The effect of radar beam angle on the beam width diameter.

**Figure 27 sensors-19-00254-f027:**
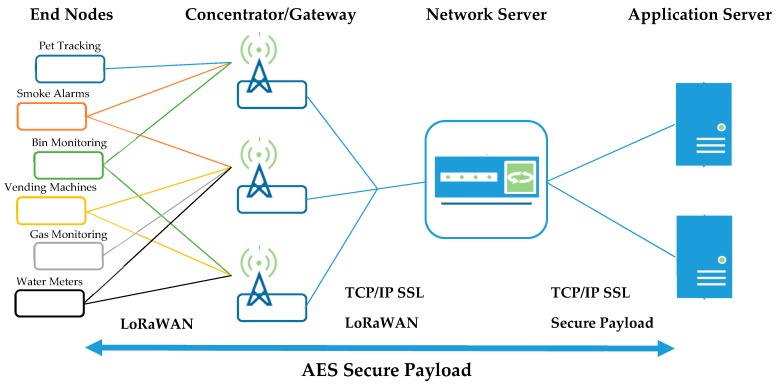
Data Packet life-cycle.

**Figure 28 sensors-19-00254-f028:**
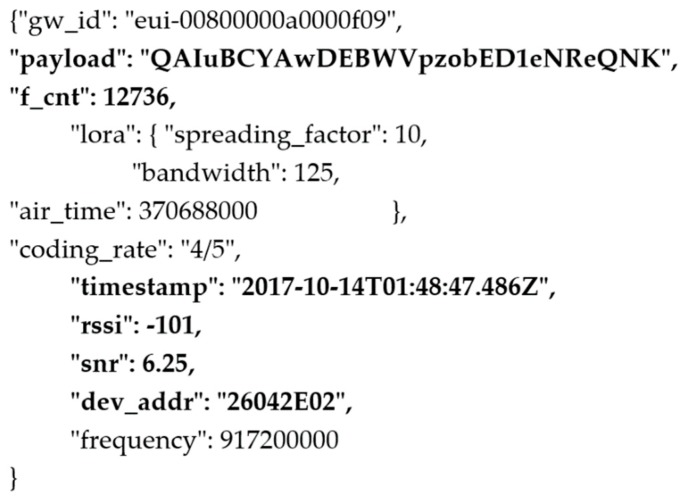
TTN Event packet capture.

**Table 1 sensors-19-00254-t001:** The effect of radar accuracy on flow measurements.

**Well Diameter (m)**	1.5	1.7	2.0	2.5	3.2	3.6
**SPS Surface Area (m^2^)**	1.767	2.269	3.142	4.909	8.042	10.179
**Flow Error (L/s)**	3.534	4.539	6.283	9.817	16.080	20.358

**Table 2 sensors-19-00254-t002:** Radio channel ranging survey statistics.

Distance (km)	Packets Received	Packets Sent	Mean SNR (dB)	Mean RSSI (dBm)	Throughput
**d < 5**	364	372	2.86	−98.73	97.8%
**5 < d < 10**	1150	1184	−0.96	−106.62	97.1%
**10 < d < 15**	1133	1200	−3.77	−107.64	94.4%
**15 < d < 20**	201	246	−3.68	−108.44	81.7%
**20 < d < 30**	50	71	−7.54	−111.80	70.4%
**d > 30**	13	27	−8.83	−113.08	48.1%

**Table 3 sensors-19-00254-t003:** Comparison of LTN technologies [22].

Property/Technology	Sigfox	OnRamp	LoRaWAN
**Network Topology:**	Star of stars	Star	Star of stars
**Channel Access Scheme:**	R-FDMA	RPMA	LoRa/CSS
**Tx Frequency Band:**	868 MHz	2.4 GHz	433/868/915 MHz
**Channel Width per Signal:**	100 Hz	1 MHz	125 KHz
**Link Data Rate (bps):**	100 bps	60–960 bps	0.3–50 K bps

**Table 4 sensors-19-00254-t004:** Data Rate and Spreading Factor vs Bitrate and Time-on-air.

LoRa Modulation Data Rate	Range	Spreading Factor	Bandwidth (kHz)	Bitrate (bps)	Receive Sensitivity (dBm)	Time-on-air & Consumption
0	7x	12	125	292	−136	22x
1	6x	11	125	537	−133	13x
2	5x	10	125	976	−132	9x
3	4x	9	125	1757	−129	7x
4	3x	8	125	3125	−126	5x
5	2x	7	125	5468	−123	3x
6	1x	7	250	10937	−120	2x
7	1x	--	--	50k	−108	1x

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
