# Peer review of "Implementation of an IoT Based Radar Sensor Network for Wastewater Management"

_sensors, 2019, doi:10.3390/s19020254_

Reviewer 1 Report

The authors conducted a very important study titled "Implementation of a Radar Sensor Network for Wastewater Management: An IoT Approach", which is good for researchers in the field of wastewater and environmental health. The whole manuscript is well written and organized. The topic is important and highly related to the journal. It is suggested that this manuscript should be accepted with some minor revisions.

The significance of the study should be addressed more in the introduction part.

As a research study, more information and references about current study should be added to strengthen the introduction part of the manuscript.

The original data should be provided as supporting materials

The caption of the table should appear above the table.

What is the water management implementations of the results?

Author Response

Thankyou for your feedback.

As per the cover letter, in rev B I have addressed the following:

Changed title to “Implementation of an IoT based Radar Sensor Network for Wastewater Management”.

Renamed wastewater distribution system to wastewater collection network (WCN).

Added 8 further research references to bolster the introduction.

Further outlined the significance of this research within the water industry.

Added the effect of temperature on sound waves/ultrasonic devices and their extra requirement for temperature compensators.

Added a flow chart figure to map the subsections for Section 3.

Fixed abbreviated words to their first appearance.

Provide more descriptive detail in regards to soft-start/stop mechanisms, and the turbulent flow caused in pump stop sequence. Propose a simple method to reduce variance of these pump types.

Moved table captions to above the tables.

Included in the Discussion some practical performance benefits already gained in-the-field by the use of these systems

Provide in the Conclusion a brief outline of the latest improvements in SCADA flow monitoring and predictive alarming, and identify where this research provides supplementary data sets to improve diurnal management tools.

Minor grammatical and reference fixes.

I hope the latest manuscript with tracked changes meet with your approval.

Regards,

Adam Drenoyanis

Reviewer 2 Report

Overall paper can be improved. Please, find my comments with attached file.

Author Response

(The authors gave the same response as above.)

Reviewer 3 Report

Brief summary

The subject of this paper is interesting for people interested in operation of wastewater collection networks, with particularly emphasis on pumping stations.

The paper presents a novel radar system to monitor the wastewater level at pumping station wells and, indirectly, estimate the pump flows. It also presents an interesting real case study about the practical implementation of a LoRa network to establish the communications from sensors spread through the network maintaining a continuous monitoring.

The paper is well written, and the references are in good number, adequate and up-to-date.

However, the structure of the paper is a bit confusing. It starts with an “Introduction”, followed by the “Methodology”, “Implementation and Results” and “Discussion”. Only after this the sections “Related Works” and “Background” present some previously published papers and some concepts related to the subject of the paper. Than the paper ends with “Conclusions” and “Future Works”.

The text is very long. Perhaps some parts could be shortened without losing important content.

In my opinion, the paper should be accepted for publication after the reorganization of the structure and the introduction of the suggestions presented below.

Specific comments

The first appearance of every abbreviation should be written in the expanded format:

… (DOL), … (HMI), … (GPS), … (RSSI), … (SNR), … (QA), … (QC), … (ACK), … (EM), … (FFT), … (ISM), … (BI)

All the other times they should be written in the abbreviated format.

²Wastewater distribution system² is not a common term, usually it is called ²wastewater collection network² or ²wastewater collection system².

“The sound waves travel at a constant speed of 340 m/s”. Is this really true? Isn’t it influenced by temperature and/or pressure?

Page 4, Lines 26-28: I didn’t appreciate the way this paragraph was written. Starts presenting three, but points just two and after, in a different sentence, adds the third.

Page 11, Lines 286-287: I don’t see the reason for this statement “This soft-start/stop mechanism creates more turbulent flow during transients”. And later, Page 12, Line 304: “This may be explained by the turbulent flows caused during transient start and stop periods”.

Page 14, Line 342: what is the meaning of ”Implementation III-A”.

Page 14, Line 352: suggest changing “SW’s” by “Shoalhaven Water”.

Page 18, Lines 43-45: I feel something is out of order in this sentence.

Author Response

Thankyou for your feedback.

As per the cover letter, in rev B I have addressed the following:

Changed title to “Implementation of an IoT based Radar Sensor Network for Wastewater Management”.

Renamed wastewater distribution system to wastewater collection network (WCN).

Added 8 further research references to bolster the introduction.

Further outlined the significance of this research within the water industry.

Added the effect of temperature on sound waves/ultrasonic devices and their extra requirement for temperature compensators.

Added a flow chart figure to map the subsections for Section 3.

Fixed abbreviated words to their first appearance.

Provide more descriptive detail in regards to soft-start/stop mechanisms, and the turbulent flow caused in pump stop sequence. Propose a simple method to reduce variance of these pump types.

Moved table captions to above the tables.

Included in the Discussion some practical performance benefits already gained in-the-field by the use of these systems

Provide in the Conclusion a brief outline of the latest improvements in SCADA flow monitoring and predictive alarming, and identify where this research provides supplementary data sets to improve diurnal management tools.

Minor grammatical and reference fixes.

In regards to the structure of the paper, we feel that the paper flows better if focussed on the presented system first before we look at the state of the art as supplemental.

I hope the latest manuscript with tracked changes meet with your approval.

Regards,

Adam Drenoyanis